# The Fabrication and Evaluation of a Capacitive Pressure Sensor Using Ru-Based Thin Film Metallic Glass with Structural Relaxation by Heat Treatment

**DOI:** 10.3390/s23239557

**Published:** 2023-12-01

**Authors:** Hodaka Otsuka, Takafumi Ninoseki, Chiemi Oka, Seiichi Hata, Junpei Sakurai

**Affiliations:** Graduate School of Engineering, Nagoya University, Nagoya 464-8603, Japan; otsuka.hodaka.m1@s.mail.nagoya-u.ac.jp (H.O.); ninoseki.takafumi@f.mbox.nagoya-u.ac.jp (T.N.); chiemi.oka@mae.nagoya-u.ac.jp (C.O.); seiichi.hata@mae.nagoya-u.ac.jp (S.H.)

**Keywords:** thin film metallic glass, MEMS, capacitive pressure sensor

## Abstract

Microelectromechanical systems (MEMS)-based capacitive pressure sensors are conventionally fabricated from diaphragms made of Si, which has a high elastic modulus that limits the control of internal stress and constrains size reduction and low-pressure measurements. Ru-based thin-film metallic glass (TFMG) exhibits a low elastic modulus, and the internal stress can be controlled by heat treatment, so it may be a suitable diaphragm material for facilitating size reduction of the sensor without performance degradation. In this study, a Ru-based TFMG was used to realize a flattened diaphragm, and structural relaxation was achieved through annealing at 310 °C for 1 h in a vacuum. The diaphragm easily deformed, even under low differential pressure, when reduced in size. A diaphragm with a diameter of 1.7 mm was then applied to successfully fabricate a capacitive pressure sensor with a sensor size of 2.4 mm^2^. The sensor exhibited a linearity of ±3.70% full scale and a sensitivity of 0.09 fF/Pa in the differential pressure range of 0–500 Pa.

## 1. Introduction

Environmental problems such as climate change and abnormal weather have become global issues [1] which can be attributed to increased emissions of greenhouse gases such as carbon dioxide, nitrogen oxides, and sulfur oxides [2]. A major source of these emissions is the exhaust produced during power generation, which varies depending on the energy source. Renewable energy sources such as solar power and wind power are environmentally friendly because they result in minimal greenhouse gas emissions. However, their wider application faces challenges such as a low power output and susceptibility to weather conditions [3,4,5]. Nuclear power is also considered environmentally friendly, but it comes with its own issues such as the need to dispose of radioactive waste and increased risk in the event of disasters [6,7]. Thus, fossil fuels remain a major contributor to power generation. Natural gas has lower greenhouse gas emissions than other fossil fuels, which has increased demand for it [8]. This has spurred the development of infrastructure for producing, transporting, and supplying natural gas, which has increased the need for sensors that can accurately measure the gas pressure.

Capacitive pressure sensors are commonly employed for natural gas applications because of their wide temperature range, high durability, low power consumption, low hysteresis, and high measurement repeatability [9,10,11,12,13,14,15,16,17,18,19,20]. These sensors measure pressure based on the change in capacitance caused by deflection of a diaphragm, which affects the distance between electrodes. However, nonlinearities may arise with increasing pressure because the capacitance is inversely proportional to the gap between electrodes. Numerous attempts have been made to improve the linearity and optimize the structure of microelectromechanical systems (MEMS)-based capacitive pressure sensors, but most studies have focused on high-pressure measurements [10,11,12,13,14,15,16,17,18]. Few studies have reported sensors designed specifically for low-pressure measurements. Of these studies, a common approach has been to increase the size or decrease the thickness of the diaphragm to increase deflection [19,20]. However, the growing demand for capacitive pressure sensors has increased the need to reduce the size and costs while improving performance. Reducing the sensor size decreases the initial capacitance and thus the deflection of the diaphragm for a given pressure. This makes it difficult to retain the change in capacitance essential for precise pressure measurement. Hence, it is necessary to augment the initial capacitance and reduce the thickness of the diaphragm.

There are two possible approaches to increasing the initial capacitance: inserting a dielectric material or reducing the gap between electrodes. Inserting a dielectric material can enhance the permittivity between the diaphragm and fixed electrode. As the diaphragm deforms, it contacts the dielectric film, and the contact area increases with the pressure, which in turn increases the capacitance. This approach is suitable for high-pressure measurements but is inapplicable to low-pressure measurements because the change in capacitance remains small until contact is established. Meanwhile, reducing the gap between electrodes may lead to them adhering to each other or to the diaphragm during the fabrication process, which requires internal stress control to reduce the diaphragm thickness. The diaphragm is conventionally made from silicon (Si), which has a high Young’s modulus that makes controlling the internal stress difficult. Thus, an alternative material to Si needs to be identified.

Thin film metallic glasses (TFMGs) have garnered much attention as a novel material for MEMS applications. TFMGs exhibit a low Young’s modulus and high elastic limit in contrast to Si [21,22,23,24], so they can undergo substantial deformation under low-pressure conditions, even when the diaphragm size is reduced. In addition, the internal stress introduced during sputtering can be managed by annealing for structural relaxation [25,26]. In a previous study, Ru-based TFMG was identified as undergoing structural relaxation by annealing below the glass transition temperature, which can decrease the introduction of tensile stress due to thermal stress when the diaphragm is flattened. Furthermore, it has a Young’s modulus of 92.7 GPa and elastic limit of 2.04%, which are suitable for use as a diaphragm in small sensors for low-pressure measurements [27]. In this study, our objective was to develop a compact sensor suitable for low-pressure measurements. It was detailed that a MEMS-based capacitive pressure sensor was fabricated by using a Ru-based TFMG as the diaphragm and evaluated the measurement performance under low-pressure conditions.

## 2. Design

Figure 1 shows the design of the MEMS-based capacitive pressure sensor, which comprised electrodes and a diaphragm. The central and reference electrodes were on a glass substrate, and the Ru-based TFMG diaphragm was on a highly doped Si substrate. The electrodes and diaphragm were bonded by a photosensitive adhesive, and the gap between the electrodes and diaphragm was controlled by the adhesive thickness. The sensor was designed to meet the target specifications of a size of 2.4 mm^2^ and maximum measurable pressure of 500 Pa. To meet the above specifications, the areas of both electrodes needed to be determined first. Figure 2 shows the configuration of the electrodes. The electrode area needed to be maximized to increase the initial capacitance. To prevent conduction during electrode deposition and keep the bonding stiffness of the adhesive layer, it was set a clearance of 0.05 mm between the central and reference electrodes and 0.35 mm from the edge of the substrate. The central electrode was circular in shape with a diameter of 1.2 mm. The reference electrode was 1.7 mm^2^ and was arranged around the central electrode with an inner diameter of 1.3 mm. The diaphragm diameter was set to 1.7 mm to increase the deflection in response to pressure. The gap between each electrode and the diaphragm was set to 2.0 µm to maximize the initial capacitance while ensuring sufficient space to avoid the diaphragm adhering to the electrode during the bonding process.

The measurable pressure range was determined from the diaphragm thickness, which affects the deflection of the diaphragm in response to pressure and in turn the change in capacitance. The diaphragm thickness was calculated by determining the upper and lower limits of the change in capacitance. The deflection of the diaphragm when gas pressure is applied can be estimated by assuming that the diaphragm is a circular disk with a fixed circumference that is subjected to a uniformly distributed load. The relationship between the radius of curvature and the deflection can be used to obtain the differential equation for the deflection at a distance *r* from the center of the diaphragm due to bending:(1)$$\frac{1}{r}\frac{d}{dr}\left[r\frac{d}{dr}\left\{\frac{1}{r}\frac{d}{dr}\left(r\frac{dw\left(r\right)}{dr}\right)\right\}\right]=\frac{p}{D}$$
where *p* is the applied pressure and *D* is the bending stiffness. The bending stiffness of the diaphragm is determined by the following formula:(2)$$D=\frac{Eh^{3}}{12\left(1-\nu^{2}\right)}$$
where *E* is Young’s modulus, *h* is the diaphragm thickness, and *ν* is Poisson’s ratio. Under the initial conditions of a zero-shear force and a finite value for the deflection, the deflection and deflection angle of the diaphragm are determined by the following formula:(3)$$w\left(r\right)=\frac{pr^{4}}{64D}+\frac{c_{1}}{4}r^{2}+c_{2}$$
(4)$$\frac{dw\left(r\right)}{dr}=\frac{pr^{3}}{16D}+\frac{c_{1}}{2}r$$

Considering the boundary conditions of zero deflection and zero deflection angle at the fixed circumference *r* = *a* of the diaphragm, it can be obtained the integration constants *c*_1_ and *c*_2_. Then, the deflection *w*(*r*) is determined by the following formula:(5)$$w\left(r\right)=\frac{\alpha {pa}^{4}}{64D}\cdot {\left(1-\frac{r^{2}}{a^{2}}\right)}^{2}$$
where *p* is the applied pressure, *a* is the diaphragm radius, and *α* is the correction factor. The values for *α*, *ν*, *E*, and *a* are already known: *α* = 9.05 × 10^−1^, *ν* = 0.33, *E* = 92.7 GPa, and *a* = 0.85 mm. This deflection is used to determine the capacitance. Because it cannot be obtained directly, it is calculated by using integration. The microelectrode area d*S* of the diaphragm at a distance *r* from the center is determined by the following formula:(6)$$dS\left(r\right)=\left\{{\left(r+dr\right)}^{2}-r^{2}\right\}\cdot \pi =2rdr\cdot \pi$$
d*r*^2^ can be ignored because it is a second-order infinitesimal term. The gap on the small area when pressure is applied is shown as *d − w*(*r*) where *d* is the initial gap. Therefore, the micro-capacitance d*C*(*r*) given to the microelectrode surface d*S* is shown using the following formula:(7)$$dC\left(r\right)=\varepsilon \frac{dS}{d-w\left(r\right)}=\varepsilon \frac{2r}{d-w\left(r\right)}dr\cdot \pi$$

This formula applies to disks or annular shapes. Therefore, the reference electrode with a square outer perimeter and circular inner frame must be approximated by a circular shape for calculation. If the areas before and after approximation are assumed equivalent, then the outer diameter *a_r_* of the annular shape can be calculated as follows:(8)$$a_{r}=\frac{L}{\sqrt{\pi }}$$
where *L* is the edge length of the reference electrode perimeter (=1.7 mm). Thus, *a_r_* was calculated as 1.92 mm. However, because the diaphragm radius was 0.85 mm, the effective area of the reference electrode ranged from an inner diameter of 1.3 mm to an outer diameter of 1.7 mm. The obtained ranges for the central and reference electrodes were used as the integration limits in Equation (7). Then, the static capacitances *C_m_* and *C_r_* when pressure is applied are shown using the following formula:(9)$$C_{m}\left(r\right)=\int_{0}^{0.6\times 10^{-3}}\varepsilon \frac{2r}{d-w\left(r\right)}dr\cdot \pi$$
(10)$$C_{r}\left(r\right)=\int_{0.65\times 10^{-3}}^{0.85\times 10^{-3}}\varepsilon \frac{2r}{d-w\left(r\right)}dr\cdot \pi$$

The change in capacitance Δ*C* is then determined by using the following formula:(11)$$\Delta C=\left(C_{m}{-C}_{m_{0}}\right)-\left(C_{r}-C_{r_{0}}\right)$$

Figure 3 shows the relationship between the deflection of the diaphragm and change in the capacitance. To ensure linearity, Δ*C* has a possible range of ≤1.22 pF. The diaphragm thickness *h* that satisfies this condition was calculated by using Equations (5) and (9)–(11), which resulted in *h* ≤ 7.5 µm. However, Equation (5) does not account for internal stress in the diaphragm, so the diaphragm thickness was set to 1.5 µm to allow for deflection.

## 3. Fabrication

### 3.1. Fabrication Process

Figure 4 shows the fabrication process of the capacitive pressure sensor, which comprises three steps: fabricating the diaphragm, fabricating the fixed electrodes, and bonding them together. To fabricate the diaphragm, a Ru-based TFMG was deposited onto a highly doped Si substrate by using a sputtering system (L-350-C; CANON ANELVA, Kawasaki, Japan) and utilizing Cr as an adhesion layer. Table 1 details the sputtering conditions. Adequately cooling the substrate helped reduce the internal stress induced by compression. The TFMG had a composition of Ru_65_Zr_30_Al_5_ (at%) [27]. Each sample was annealed under a high vacuum to relieve the initial internal stress and introduce tension. The diaphragm was fabricated by deep reactive ion etching (DRIE) of the Si substrate underneath the TFMG.

For the fixed electrodes, through-holes with a diameter of φ0.3 mm were made to connect the wiring between the upper and lower portions. Then, the electrodes were fabricated by depositing Au onto a glass substrate using a sputtering system (E-200S; CANON ANELVA, Kawasaki, Japan) and utilizing Cr as an adhesion layer. The backside of the substrate was also sputtered with Cr and Au to fabricate the electrode pad and wiring. Table 2 details the sputtering conditions.

Finally, the diaphragm and fixed electrodes were joined by an indirect bonding method using an adhesive. This method was chosen because the common method of anodic bonding to join glass and Si requires the application of high voltage and a temperature of over 200 °C. In contrast, the indirect bonding method induces adhesion of the diaphragm due to compressive stress caused by electrostatic attraction and thermal stress. A photosensitive adhesive (TMMR SA390N; Tokyo Ohka Kogyo, Kawasaki, Japan) was used to bond the diaphragm and fixed electrodes, which were bonded at low temperatures of 80–110 °C and could be patterned by lithography without introducing much thermal stress. After the adhesive was applied to the fixed electrodes by spin-coating and was patterned, the diaphragm was bonded to the electrode by heating it to 110 °C under a pressure of 0.2 MPa for 2 min. Then, the bonded diaphragm and fixed electrodes were annealed on a hot plate at 180 °C for 90 min. They were then cut into chips to fabricate capacitive pressure sensors.

### 3.2. Annealing Conditions

In a previous study, the fabricated TFMG diaphragm became a dome because of compressive stress introduced during sputtering. To flatten the diaphragm, the initial compressive stress was reduced by annealing utilizing β relaxation [28]. Figure 5 shows an optical microscopy image and the cross-sectional profile of the backside of the diaphragm after annealing at 250 °C for 1 h. A flat diaphragm was successfully fabricated. Figure 6 shows the bulge test. Samples were prepared by using the same fabrication process as shown in Figure 4, but the temperature during annealing was varied while the annealing time was kept constant at 1 h. After a sample was set in the base jig, a stainless-steel holding jig was used to secure the Si substrate from the top, and the surroundings were covered with polyimide tape to prevent pressure from leaking and to ensure accurate measurement. A pressure calibrator was connected to the hose, and pressure was applied to the diaphragm. The pressure–deflection characteristics were evaluated by measuring the deflection of the diaphragm using white-light interference. Controlling the deflection of the diaphragm according to the annealing temperature allowed the change in capacitance to be regulated. When the change in capacitance exceeds 20% of the initial capacitance, the linearity of the capacitance–deflection relationship is lost [28]. Therefore, the conditions under which the deflection of the diaphragm was less than 0.7 µm at the upper limit of the pressure needed to be determined.

Figure 7 shows the deflection characteristics of the diaphragm at each annealing temperature. The deflection increased at lower annealing temperatures for a given pressure. This is because increasing the temperature increased the tensile stress introduced into the diaphragm by thermal stress during cooling, which made it difficult for the diaphragm to deform. The annealing conditions were set to 310 °C for 1 h, which resulted in a deflection of 0.6 µm when a differential pressure of 500 Pa was applied.

### 3.3. Fabrication Results

Twenty-five MEMS-based capacitive pressure sensors were fabricated from one lot, of which 21 were found to be viable. Figure 8a,b show optical microscopy images of the top and bottom of a sensor. No damage was observed for the diaphragm, but a fringe pattern was observed. Figure 8c shows scanning electron microscopy (SEM) images of the cross-section. The adhesive thickness ranged from 3.3 µm to 7 µm, which may be attributed to insufficient pressure during bonding, the surface roughness of the glass, and a non-uniform thickness of the adhesive layer.

## 4. Performance Evaluation

### 4.1. Evaluation Method

Figure 9 shows the setup for measuring the capacitance of the pressure sensor. To prevent parasitic capacitance during measurements, the jig for the mounted sensor was fabricated using fused deposition modeling 3D printing from ABS resin. Conductive tape was attached to the top of the jig to extract electrodes from the diaphragm via the highly doped Si substrate. The insulating tape was attached to the opposite side for angle correction. A liquid gasket was applied around the sensor to prevent gas leakage. The capacitance of the pressure sensor was measured by using an inductance–capacitance–resistance (LCR) meter at a measurement frequency of 100 kHz and applied voltage of 1.0 V. The LCR meter was connected to the wiring pads and conductive tape of the pressure sensor by conductive wires. There were two wiring pads, each connected to the central electrode and reference electrode. The applied pressure was increased from 0 Pa to 500 Pa in increments of 100 Pa and was then decreased to 0 Pa. The capacitance was measured at both the central and reference electrodes.

### 4.2. Evaluation Results

Figure 10 shows the initial capacitances of the 21 pressure sensors (i.e., Chips 1–21). Both the central and reference electrodes exhibited variations in the range of 0.5–4 pF. The gap between electrodes was calculated from the initial capacitance to be a maximum of 9 µm, which was 4.5 times greater than the design value of 2 µm. This was attributed to the lack of control over the film thickness of the photosensitive adhesive by spin-coating, which increased the variability in the thickness of the photosensitive adhesive and resulted in incomplete bonding across the entire substrate. Table 3 presents the measured and designed values of the initial capacitance and estimated gap of Chip 1, which had the highest initial capacitances at the electrodes among the fabricated sensors. Although the designed initial capacitances of the center and reference electrodes were 5.00 and 6.91 pF, respectively, the measured initial capacitances were 2.08 and 3.90 pF, respectively, which were approximately half the design values. The areas of the central and reference electrodes were 1.13 and 1.56 mm^2^, respectively. The estimated gaps for the central and reference electrodes were 4.82 and 3.53 µm, respectively. The variability in the adhesive thickness indicates that the gap may not have been constant.

Figure 11 shows the relationship between the capacitance and applied pressure at the central electrode of Chip 1. The pressure was increased in increments of 100 Pa from 0 to 500 Pa and then decreased back to 0 Pa. This cycle was repeated 10 times. Figure 11 shows the data for the 10th cycle. As the applied pressure increased, the capacitance increased linearly. As the pressure decreased, the capacitance also linearly decreased. Thus, there was no hysteresis. The linearity was within ±3.70% full scale (F.S.), and the hysteresis was 2% F.S. The sensitivity of Chip 1 was 0.09 fF/Pa for a pressure range of 0–500 Pa, which is much lower than the design sensitivity of 0.31 fF/Pa. Figure 12 shows the relationship between the capacitance and applied pressure at the reference electrode of Chip 1. The capacitance remained almost constant regardless of the change in pressure, which indicates that the reference electrode was working well.

The deflection of the diaphragm in Chip 1 from the capacitance of the central electrode by using Equation (9) was calculated. Figure 13 shows the estimated deflection of Chip 1, which was less than the measured deflection in the bulge test. This can be attributed to the variation in the initial internal stress induced by the sputtering process and insufficient stress relaxation. The stress relaxation progressed more for the MEMS sensor than during the bulge test, so a large tensile stress was applied during cooling. Therefore, the annealing conditions need to be optimized so that the initial internal stress can be relieved at a low temperature over a long time.

## 5. Conclusions

A Ru-based TMFG was used to fabricate a capacitive pressure sensor and its performance was evaluated. By adequately cooling the substrate during sputtering, the initial internal stress could be mitigated and tensile stress was introduced at a low annealing temperature. Annealing conditions were set to realize a deflection of 0.6 µm at a pressure of 500 Pa. Utilizing a photosensitive adhesive to bond the diaphragm and fixed electrode reduced the amount of thermal stress introduced to prevent sticking. The results showed that a pressure sensor with a flat diaphragm structure that demonstrated a sensitivity of 0.09 fF/Pa, linearity of ±3.70% F.S., and hysteresis of 2% F.S. in the measurement range of 0–500 Pa was successfully fabricated. However, the sensor performance deviated from the designed characteristics, which was attributed to variation in the induced internal stress during sputtering and the annealing conditions being insufficient to relax the initial internal stress.

Although the sensor performance was not satisfactory, the results indicate that the fabrication of a capacitive pressure sensor from a TFMG is feasible. In future works, an attempt to improve the sensor performance will be made by determining suitable annealing conditions to achieve sufficient stress relaxation and reviewing the bonding process to control the gap size.

## Figures and Tables

**Figure 1 sensors-23-09557-f001:**
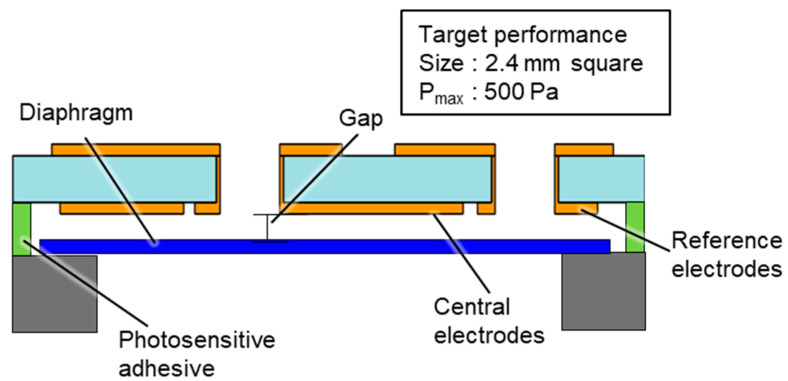
Structure of the capacitive pressure sensor.

**Figure 2 sensors-23-09557-f002:**
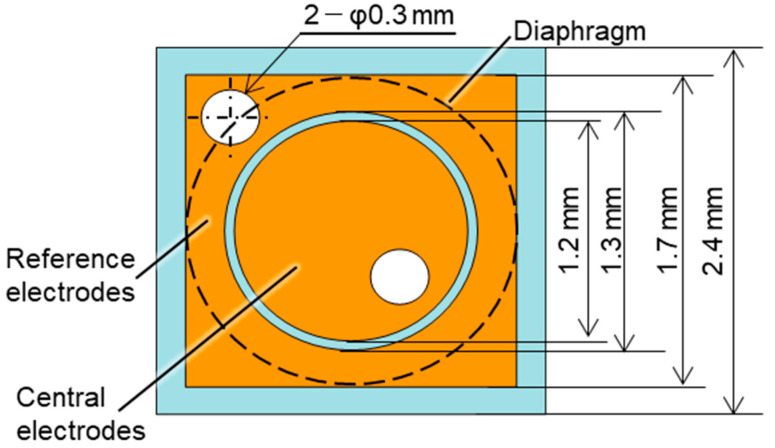
Electrode configuration.

**Figure 3 sensors-23-09557-f003:**
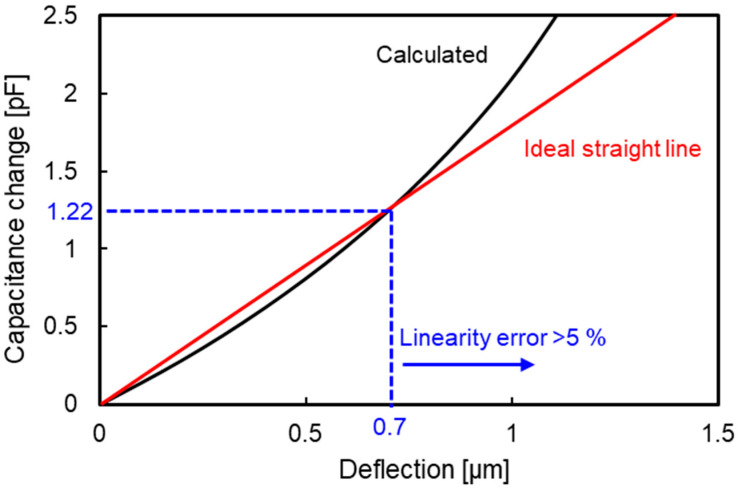
Relationship between the deflection of the diaphragm and change in capacitance.

**Figure 4 sensors-23-09557-f004:**
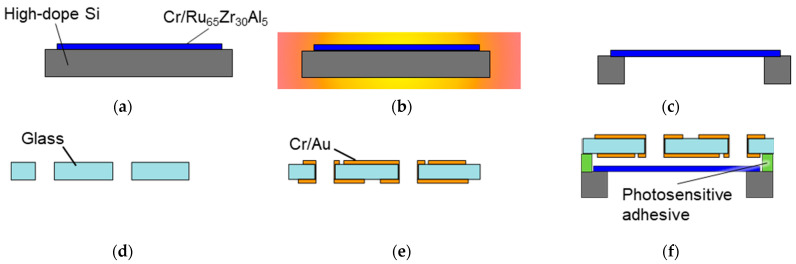
Fabrication process of the capacitive pressure sensor: (**a**) sputtering the TFMG, (**b**) annealing, (**c**) DRIE, (**d**) milling, (**e**) sputtering Au, and (**f**) bonding.

**Figure 5 sensors-23-09557-f005:**
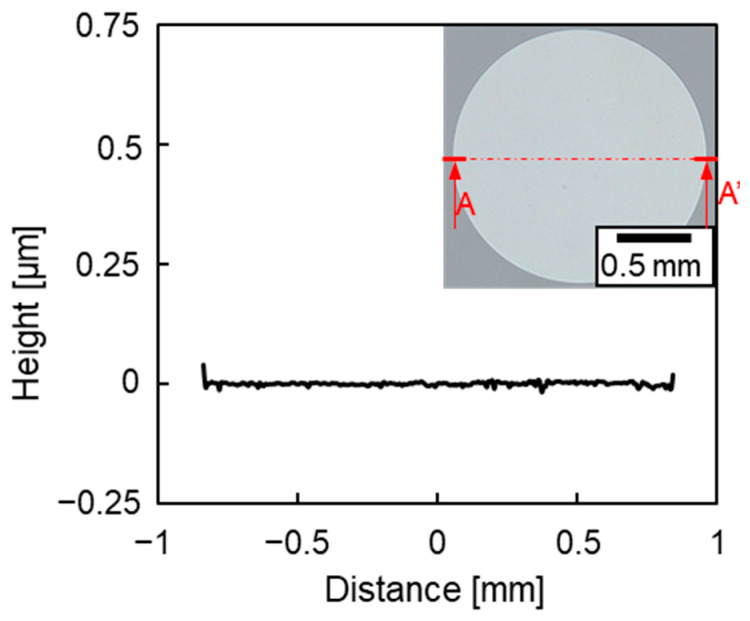
A–A′ cross-section of the diaphragm (inset) and its profile.

**Figure 6 sensors-23-09557-f006:**
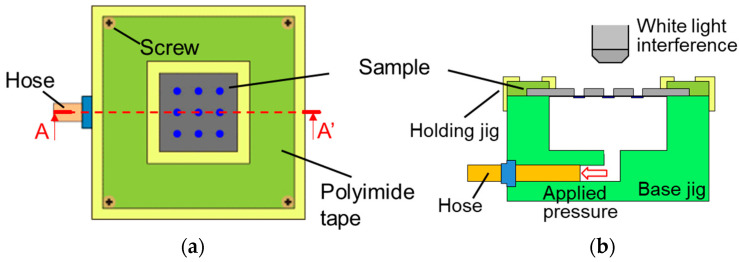
Bulge test: (**a**) top view and (**b**) A–A′ cross-section.

**Figure 7 sensors-23-09557-f007:**
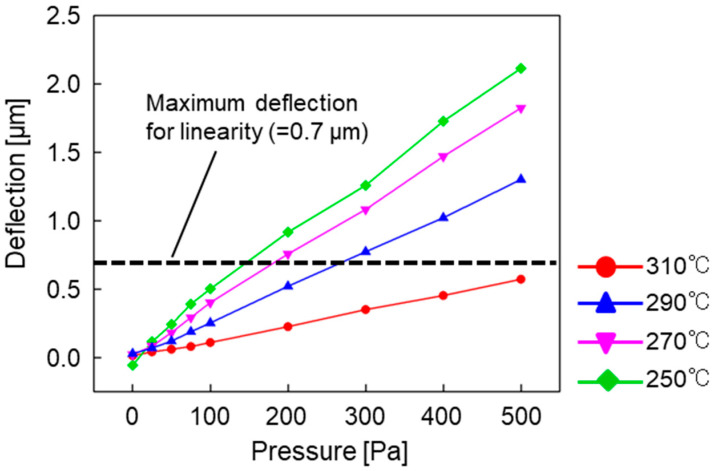
Dependence of the annealing temperature on the pressure–deflection characteristics.

**Figure 8 sensors-23-09557-f008:**
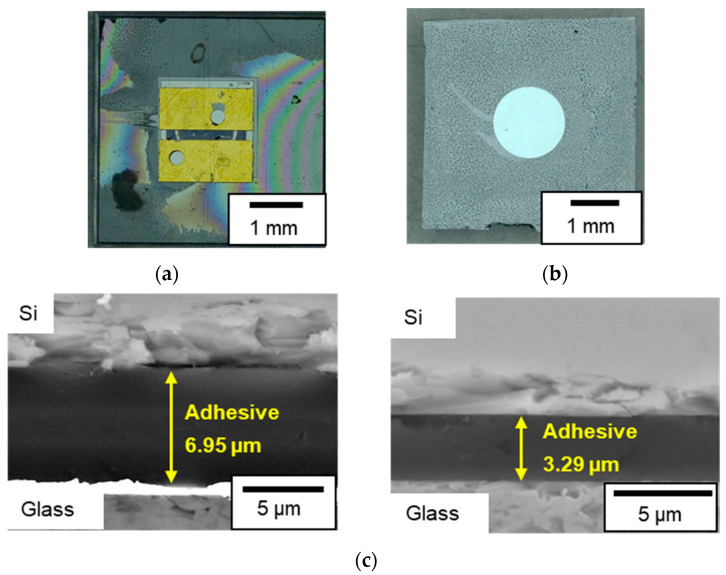
Fabricated pressure sensor: (**a**) top view, (**b**) bottom view, and (**c**) SEM image of the cross-section.

**Figure 9 sensors-23-09557-f009:**
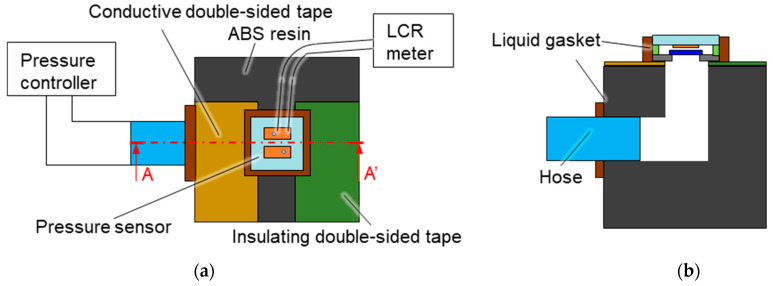
System for measuring the capacitance of the pressure sensor: (**a**) top view and (**b**) A–A′ cross-section.

**Figure 10 sensors-23-09557-f010:**
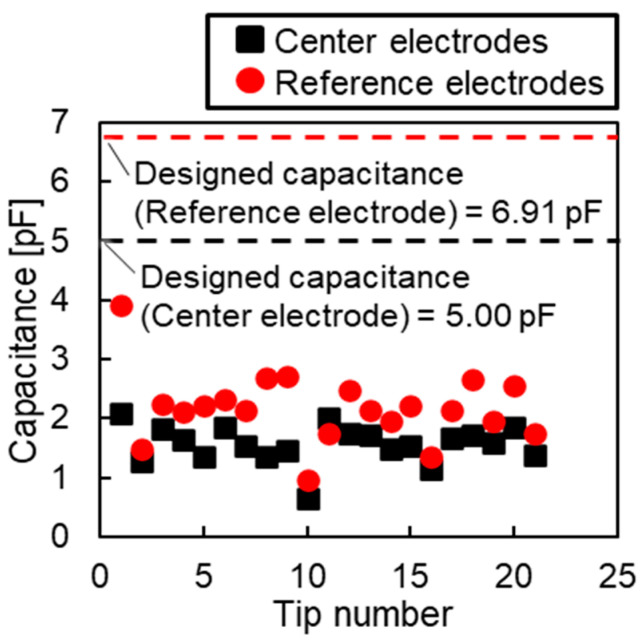
Measured initial capacitances of fabricated sensors.

**Figure 11 sensors-23-09557-f011:**
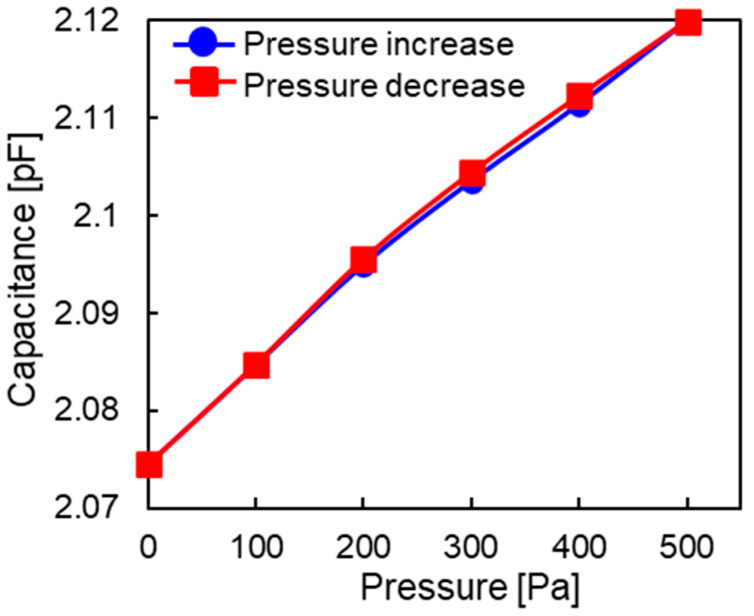
Relationship between capacitance and applied pressure at the central electrode of Chip 1.

**Figure 12 sensors-23-09557-f012:**
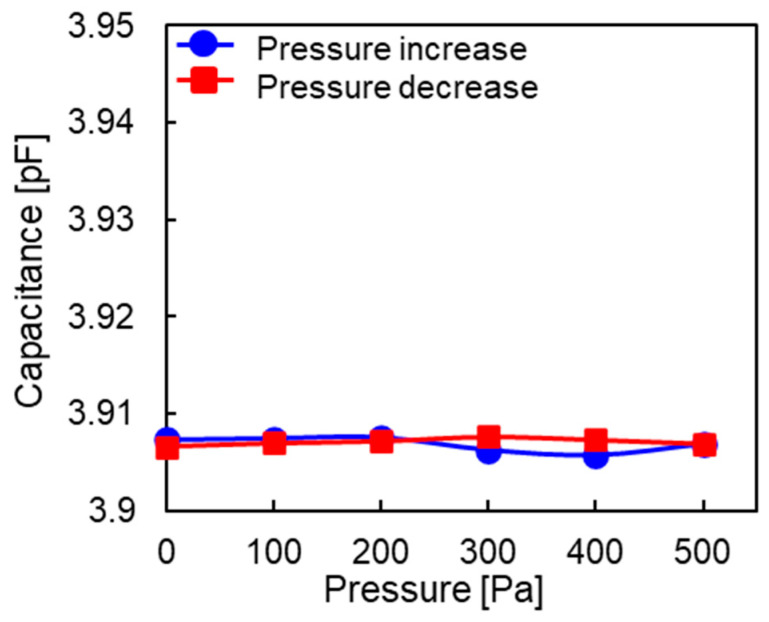
Relationship between the capacitance and applied pressure at the reference electrode of Chip 1.

**Figure 13 sensors-23-09557-f013:**
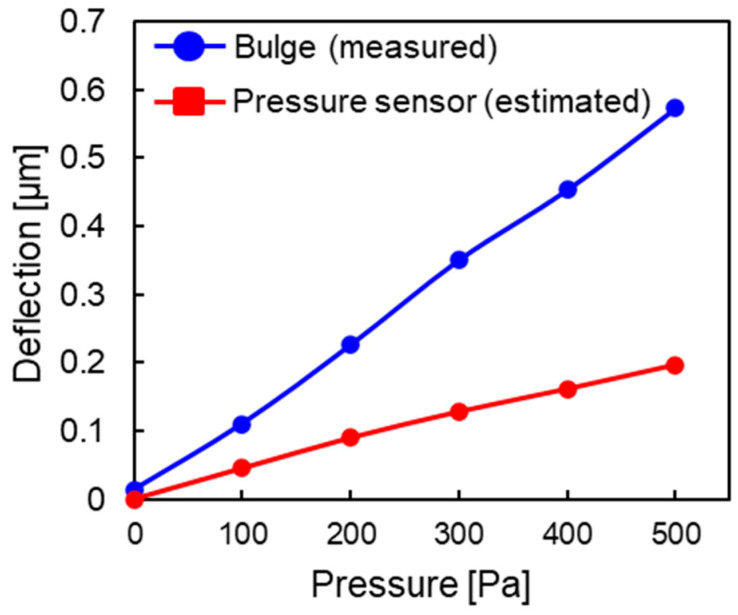
Pressure–deflection characteristics of the pressure sensor compared using the bulge test.

**Table 1 sensors-23-09557-t001:** Experimental conditions of sputtering Ru-based TFMG.

Target	Ru_65_Zr_30_Al_5_	Cr
Ar pressure [Pa]	0.8	0.5
RF power [W]	100	100
Time [min]	40	1

**Table 2 sensors-23-09557-t002:** Experimental conditions for sputtering Au.

Target	Au	Cr
Ar pressure [Pa]	1.0	1.0
RF power [W]	200	200
Time [min]	8	1

**Table 3 sensors-23-09557-t003:** Measured and designed values of the initial capacitance and estimated gap for Chip 1.

Chip	Central Electrode	Reference Electrode
Designed initial capacitance [pF]	5.00	6.91
Measured initial capacitance [pF]	2.08	3.90
Estimated gap [µm]	4.82	3.53

## Data Availability

Data are contained within the article.

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
