# Peer review of "The Fabrication and Evaluation of a Capacitive Pressure Sensor Using Ru-Based Thin Film Metallic Glass with Structural Relaxation by Heat Treatment"

_sensors, 2023, doi:10.3390/s23239557_

Round 1
Reviewer 1 Report
Comments and Suggestions for Authors
Dear authors, please check the attached file with my comments.

Reviewer 2 Report
Comments and Suggestions for Authors
1. English revision by native spaker is needed.
2. Check typoes (for example, 'date' in line 273)
2. More backgraound shoud be provieded in the introduction. Why did you choose Ru-based TFMG?
3. The detail of the TFMG deposition should be provided .
4. More information about the bulge test should be provided.
5. Performance comparison with the sensor made of other materials or commercial sensor is needed.
Comments on the Quality of English LanguageEnglish revision by native spaker is needed.
Reviewer 3 Report
Comments and Suggestions for Authors
Point 1. The article must be formulated as an unitary material. Also, the paper should be written in the impersonal mode (we fabricated should be replaced with it was fabricated).
Point 2. The Abstract must be rewritten. Place the subject in a broad context, highlight the purpose of the study and strengthen the conclusions.
Point 3. The English must be revised throughout the work. Bellow, the authors can find some examples:
Lines 10 and 70 must be reformulated, suggestion: In this paper, it was describe/detailed the fabrication
articulation of words (lines: ...205, 210, 251, 269, 270....);
rephrasing (lines 41, 152, 168, 176, 182, 211, 223, 254, 293)
replacing expressed with shown (lines: 138, 139, 157)
given replaced with determined (lines: 106, 116)
In the sentence on the line 224 there are 2 figures 8(a) and (b)
Lines 252-253, the reason is not understood
Line 275, must be reformulated, add the value obtained
Line 306 , the verb is missing.
Point 4. Improve the introduction by highlighting the degree of novelty that the study brings. Also, add recent references, enhance the state of the art with recent arguments related to the material used and the conditions under which the sensor responds
Point 5. The figures should be placed right below the text in which they were cited first (eg. Fig. 1 must be placed after line 79...).
Point 6. The authors need to expand the conclusions, highlighting the importance of what it was obtained and the degree of novelty and originality.
Point 7. Apart from the improvement of the sensor’s sensitivity compared to the one published in IEEE Proceeding, 2018 (ref. 15), I recommend the authors to explain the originality of the present work both in the Abstract and in the Evaluation of pressure sensors section.
Comments on the Quality of English LanguagePoint 3. The English must be revised throughout the work. Bellow, the authors can find some examples:
.Lines 10 and 70 must be reformulated, suggestion: In this paper, it was describe/detailed the fabrication
articulation of words (lines: ...205, 210, 251, 269, 270....);
rephrasing (lines 41, 152, 168, 176, 182, 211, 223, 254, 293)
replacing expressed with shown (lines: 138, 139, 157)
given replaced with determined (lines: 106, 116)
In the sentence on the line 224 there are 2 figures 8(a) and (b)
Lines 252-253, the reason is not understood
Line 275, must be reformulated, add the value obtained
Line 306 , the verb is missing.
Round 2
Reviewer 3 Report
Comments and Suggestions for Authors
I accept the publication of the article in its present form.
Best regards!